# OpenReview forum: "GRADIEND: Feature Learning within Neural Networks Exemplified through Biases"
_ICLR.cc/2026/Conference — ICLR 2026 Poster_

### Official Review · Reviewer_43sP · 2025-10-30

**Soundness:** 3
**Presentation:** 2
**Contribution:** 4
**Rating:** 8
**Confidence:** 4

**Summary:**

This paper proposes to utilize the internal representations of a Language Model (LM) to analyze and address biases embedded within its parametric knowledge.

The core methodology involves training an encoder-only LM, wherein tokens representing target interpretations or attributes are systematically masked. During this training phase, gradients are meticulously collected and subsequently analyzed to compute orthogonal directions that effectively decouple feature updates associated with biases from those of non-biased information. Finally, these derived orthogonal directions are intended to be deployed to mitigate biases during the actual training process of the larger Language Models.

**Strengths:**

1. This paper introduces a highly effective methodology for analyzing and potentially mitigating representational biases withi language models. The proposed method leverages the gradients collected during the training process of the target LM. Specifically, it employs a single neuron bottleneck encoder-decoder network to classify updates into bias-related and non-bias-related feature classes. The experimental results demonstrate that this approach is robust and significantly useful in isolating these components.
2. The innovative application of the encoder-decoder network extends beyond classification to bias mitigation, which is a particularly nuanced and valuable contribution. The network is trained by reconstructing the gradients of both bias and non-bias updates. By strategically modifying the decoder component, the approach can effectively modulate the model's propensity to learn or reinforce undesirable bias information during subsequent training.
3. This line of research is highly pertinent and contributes significantly to the current landscape of large model safety and alignment. Given the demonstrable harm caused when models inadvertently internalize and propagate toxic information or biased distributions found in real-world data, the work presented here is profoundly meaningful. In my assessment, studies focusing on the technical mechanisms of bias identification and mitigation, as detailed in this paper, are essential for promoting responsible AI development.

**Weaknesses:**

Given that fine-tuning remains the dominant paradigm in the modern development and deployment of Language Models (LMs), the current methodology presented in this paper appears to overlook its direct applicability within this context.

It would significantly strengthen the paper's relevance and impact to include a dedicated discussion on how the proposed method can be practically leveraged or adapted during the fine-tuning process. This discussion should address potential complexities, necessary modifications to the core technique, or provide a theoretical justification for its utility in a fine-tuning environment.

**Questions:**

1. The current description lacks sufficient detail regarding the gradient collection and utilization process, leading to confusion about the method's implementation. Specifically:

- Gradient Acquisition: It is unclear precisely how the gradients are collected in the proposed method. The paper must specify which layers (e.g., embedding, attention block, or final layer) are targeted for gradient emphasis and extraction, as this significantly impacts the captured feature information.

- Training Set Construction: The process of using the gradient differences to train the bottleneck encoder-decoder network is not clearly delineated. Authors must elaborate on the strategy used to split the collected gradients (or gradient differences) into appropriate training and validation sets for the reconstruction task.

2. The design choice for the bottleneck module warrants further discussion and justification.

- Dimensionality Analysis: Could the authors explore the impact of adding more neurons (i.e., increasing the dimensionality) to the bottleneck module? While a smaller bottleneck enforces stronger compression, it may limit the representational capacity needed to distinctly separate the bias and non-bias classes.

- Classification Performance: A detailed analysis or ablation study should be provided to demonstrate whether a slight increase in bottleneck size could lead to improved classification performance and better feature discrimination between the different update types.

**Details Of Ethics Concerns:**

N / A

---

> ### Author Response · Authors · 2025-11-21
> **Answer to Reviewer 43sP**
>
> Thank you for the review. Please see our general comment for a summary of the main revisions made in response to all reviews.
> All references to sections, tables, etc. in our answer correspond to the revised manuscript unless stated otherwise.
>
> **Questions:**
>
> **1. Method Details:**
>
> - *Gradient Acquisition:* We briefly mention in Section 5.1 that we use all layers except for the token prediction layers of the base models (lines 236-237 in the original paper). We made this now more explicit when describing the training in Section 5.1. We follow this approach to allow GRADIEND to learn by itself which weights are most informative, instead of arbitarily selecting specific layers, as done in other works using, e.g., SAEs [1, 2]. We now explicitly point out this difference when discussing SAEs in Section 2.1. Understanding which neurons (e.g., layers, aspects of the transformer, etc.) are used by GRADIEND is an aspect for future work, where the insights from GRADIEND will be used to study the internal structure of neural networks, something that is not (as easily) possible with approaches that a priori select a subset of neurons.
> - *Training Set Construction:* For gender, both the GENTER template texts (containing a [NAME] and [PRONOUN] template key) and the NAMExact names are randomly split into train/validation/test sets (Appendix D.4). During GRADIEND training, only the train templates with train names are used, and likewise for validation and test, ensuring porper separation. For race and religion, the GRADIEND data consists only of a single template key of bias target words (e.g., Asian or Jewish), hence simple data splits are sufficient.
>
>
> **2. Design Choices:**
>
> We assume that the classification performance here refers to using the feature neurons, i.e., the bottleneck between encoder and decoder to determine the class of the feature, e.g., the gender. While we acknowledge that using more neurons here could possibly improve this, this would be detrimental towards the key strengths of GRADIEND: the neurons would possibly not have such a well-defined meaning that directly reflects the *intended* encoding of the feature. Instead, one would have to run additional analysis regarding the specific meaning of each feature neuron and could, possibly, subsequently use them to re-write models in the same way we did. By using strictly constrained bottlenecks with clearly defined expectations for the learned features, we avoid this. Nevertheless, we agree that using a single feature neuron as we do enforces stronger compressions, and that using more feature neurons could improve debiasing, and we now explicitly mention this in Section 6 (Limitations and Open Questions).
>
>
> **Weakness: GRADIEND and Fine-Tuning**
>
> While this study only demonstrated the application of GRADIEND on pre-trained models, it can also be applied to finetuned ones. We add such a discussion in the new Appendix H, where we apply GRADIEND before and/or after fine-tuning the SuperGLUE WSC task. Our analysis shows that the debiasing effect typically only changes slightly whether GRADIEND is applied before and/or after the fine-tuning.
>
>
> [1] Bricken, Trenton, et al. “Towards Monosemanticity: Decomposing Language Models With Dictionary Learning.” Transformer Circuits Thread, Anthropic, Oct. 4, 2023.
>
> [2] Brinkmann et al., 2025. Large Language Models Share Representations of Latent Grammatical Concepts Across Typologically Diverse Languages. In NAACL-HLT 2025.

---

> > ### Comment · Reviewer_43sP · 2025-11-22
> >
> > Appreciate your detailed reply. There are clarifications and some more questions for you.
> >
> >
> >
> > **Clarification**
> >
> > As for the question before, I'm actually asking: Can we use the gradients in the fine-tuning process to conduct debiasing? Because it is really labor-intensive to post-train a large model.
> >
> >
> >
> > **Question**
> >
> > In your reply, you mentioned that the training, validation and test set are randomly split, which does not guarantee a sufficient difference between the samples of the training set, validation set, and test set. It is strongly recommended that you introduce a new test set that is totally different from the other data, such as in format, content, sequence order, and language. This could empirically show the generalization of your method.

---

> > > ### Author Response · Authors · 2025-11-25
> > > **Answer to Reviewer 43sP**
> > >
> > > Thank you for your quick response.
> > >
> > > **Clarification: Using Fine-Tuning for GRADIEND:**
> > >
> > > We assume you mean the question as follows: *Can the gradients produced during fine-tuning be directly reused to train a GRADIEND model, which can be used for debiasing?*
> > >
> > > Our answer is no. The gradients obtained during fine-tuning do not contain the factual-counterfactual signal that GRADIEND requires. GRADIEND relies on gradients along an explicitly constructed orthogonal (debiasing) axis, which does not arise in typical task fine-tuning.
> > >
> > > A second possible interpretation is whether fine-tuning *itself* can debias a model, which indeed is possible, for instance through training on a counterfactually augmented or otherwise balanced dataset. This is indeed a known approach in literature, but it follows a different mechanism than GRADIEND: the debiasing comes from modifying the data distribution rather than extracting an explicit (de)bias direction.
> > >
> > > Nevertheless, our fine-tuning analysis (Appendix H) shows that GRADIEND's debiasing effect typically persists after fine-tuning, indicating that applying GRADIEND once, before fine-tuning the debiased model on multiple tasks, is feasible in practice.
> > >
> > >
> > > **Additional Experiment:**
> > >
> > > We agree that demonstrating generalization on data that differs substantially from the training data is important. Given the very short rebuttal timelines and our mostly monolingual model setting, we could not perform a large multi-lingual or multi-format evaluation as suggested. However, we conducted a focused experiment that meets the core requirement of testing the encoder on structurally different data. The new analysis is now included in Appendix E.3 (with data generation details in Appendix B.7). We also point out the experiments in the main part at lines 253-254.
> > >
> > > We evaluate the GRADIEND encoder on a subset from English Wikipedia. Our derived dataset (WikiGender) differs from BookCorpus (used for training) in several key aspects:
> > >
> > > - **Format:** Wikipedia contains headings and multi-sentence paragraphs, unlike the single-sentence BookCorpus templates used in training.
> > > - **Content:** Wikipedia covers encyclopedic definitions and descriptions rather than narrative book content.
> > > - **Target Terms:** Instead of only masking *she/he* (as in training), we additionally evaluate on other gendered word pairs (e.g., *woman/man* and *daughter/son*).
> > >
> > > ***Summary of Findings:***
> > >
> > > - The *she/he* encoded values generalize well to WikiGender for all models.
> > > - For BERT base, BERT large, and DistilBERT, the gender encoding generalizes also to other gendered terms, though being slightly noisier than the *she/he* encodings. This is expected because the other pairs are more general and allow for more alternatives in the language modeling.
> > > - RoBERTa behaves differently, as it appears to encode a narrow *she/he* feature rather than a general gender feature. This gives us a new insight that shows that GRADIEND can possibly overfit to the specific counterfactual (*she/he*) instead of the underlying feature.
> > > - For GPT-2 and LLaMA, generalization to non-*she/he* pairs is weaker but still present. This is expected because the more general concepts are harder to strictly identify if only the context left of the word is available.
> > > - Llama-Instruct shows a consistent shift towards male-associated values for non-*she/he* pairs.

---

### Official Review · Reviewer_jNB4 · 2025-10-31

**Soundness:** 3
**Presentation:** 2
**Contribution:** 3
**Rating:** 4
**Confidence:** 3

**Summary:**

This study proposes a new technique that uses model gradients to learn a feature encoding for societal topics like gender, race, and religion. The authors show that this encoding can be used to adjust the model's bias. The paper includes many experiments on different models. The results show that the method has a decent effect on changing model bias while keeping its capabilities.

**Strengths:**

The main strength is the new technique. Using gradients to learn a feature for bias is an interesting idea. The study also has a wide range of experiments. The authors evaluated on a vast set of models, which is good. The results show the method can change the models, which is a high impact.

**Weaknesses:**

- The paper's primary weakness is its presentation, which makes the methodology difficult to understand. The authors first explain the methodology with formal definitions and mathematics before providing a high-level overview. In its current format it is really hard for the reader to comprehend what you are actually doing. The paper would be significantly improved by first explaining the method conceptually and providing intuition about each step, and then diving into the formal definitions. An example after your formal definition would further improve the comprehensiveness.

- This lack of clarity extends to the experimental setup. As an experimental focused study, it fails to adequately give intuition about different parts of the methodology. For example, the creation of the data for racial and religious bias contexts is unclear (Lines 201-202). For gender, it is questionable whether the name-pronoun data (Lines 198-199) is enough to capture gender information, or if other contexts like gendered nouns or adjectives are needed.

- Methodologically, I am not convinced that this method is as effective for generative models. The approach seems more focused on encoder models. For generative models, the feature neuron is constructed from a partial section of the data (the left-side context), which may be insufficient to learn a robust feature representation.

- Finally, the evaluation of the results is not fully convincing. Looking at the individual outputs of the models in table 12, does not strongly indicate the superiority of this method. The aggregated results in Table 2 are used to make stronger claims, but the paper does not specify the methodology in detail on how you did calculate these aggregated ranks.

Minor presentation issues also detract from the paper:

- The referencing of (H1) and (H2) in lines 185-187 and 255 is inconvenient. The phrases should be written out. In Line 255, it is unclear if (H1) is part of the sentence.

- In Line 146, the manuscript uses incorrect starting quotation characters

**Questions:**

- What if the target words are not a single token? How does your methodology account for this?

- How did you ensure that there is enough information for the generative models in the left side of the masked token to compute meaningful gradients?

- Why did you present StereoSet SS and LMS scores separately? Given that you are measuring bias and model capabilities. Why you did not present the ICAT score?

---

> ### Author Response · Authors · 2025-11-21
> **Answer to Reviewer jNB4**
>
> Thank you for the review. Please see our general comment for a summary of the main revisions made in response to all reviews.
> All references to sections, tables, etc. in our answer correspond to the revised manuscript unless stated otherwise.
>
> **Questions:**
>
> - **Handling multi-token targets:** For encoder-only models, multi-token targets are naturally supported by inserting one `[MASK]` token per target token. The MLM loss then accounts for all target tokens. We extended our description in the paper, see Appendix D.2. For decoder-only models and the CLM objective, this is a current limitation that we acknowledge. This is discussed in the Limitations (Section 8) and further elaborated in Appendix D.3. We agree that extending the framework to multi-token targets is an important future work. One straightforward option is an iterative approach that computes single-token gradients for each step and averages them to derive inputs for GRADIEND training. We now mention this potential approach explicitly in Section 6.
> - **Left Context of Decoder-Only Models:** For gender, the data construction (Appendix B.3) ensures sufficient left-context information before the mask, as a name always occurs before the masked pronoun (which relates to the name). For race and religion, we do not have such constraints, i.e., some samples may provide limited contextual information. These cases introduce noise, but our empirical results show that GRADIEND still learns a meaningful feature neuron and bias-changing decoder representation.
> - **SS and LMS:** Our evaluation of GRADIEND-modified models has two distinct goals: quantifying bias (SEAT and SS), and ensuring that model capabilities are preserved (LMS, GLUE, and SuperGLUE). Because each goal has multiple metrics, we report them separately. ICAT is an aggregation of SS and LMS and does not add additional insights.
>
> **Weaknesses:**
>
> - **Presentation:** We acknowledge that the description of GRADIEND is mostly based on the mathematical formalism. However, we note that we summarize GRADIEND with the desired high-level overview in Figure 1 to illustrate our approach, including an example. While we would prefer to provide more examples, we are, unfortunately, constrained in the length of the paper.
> - **Clarity Experimental Setup:**
>     - *Race and Religion Data:* We acknowledge that the data generation is described only very briefly in the main part of the paper due to space constraints. However, more information can be found in Appendix B (Data) as mentioned at the end of the main part data section (Section 4). Specifically, Appendix B.4 adds full generation details on race and religion data generation. To further enhance clarity in the main part of the paper, we now provide more generation details for race and religion data in Section 4.
>     - *Gender Data:* Our results demonstrate that name–pronoun pairs capture the gender feature sufficiently for both encoding and modifying gender bias. We agree that including adjectives or nouns could potentially enrich the signal, but this is out of scope for our study. We now briefly discuss this aspect in Appendix B.3 (Details of Gender Data).
> - **Effectiveness on Decoder-only Models:** We agree that the effectiveness of decoder-only models trained based on CLM naturally depends on the quality of left-context information. However, for gender, our encoding analysis (Table 1) shows that the decoder-only models outperform all encoder-only models in Cor_Enc. This demonstrates that GRADIEND is not limited to encoder-only models, but also works well with generative decoder-only models. The weaker encoding results of decoder-only models (mostly Llama-based models) for race and religion probably stem from the multi-token targets used in these settings and models (and that they are not yet supported), as well as the noisier data.
> - **Convinceness of Debiasing Evaluation:** We expanded Appendix G.2 with a detailed explanation of how the mean proportional ranks in Table 2 are computed, i.e., based on the raw results shown in Table 14-17 (note that the table numbers refer to the updated paper; Table *12* of the first submission is now Table *14*!). Analyzing the raw tables indeed shows that GRADIEND is not always the best (weight-changing) approach, e.g., CDA outperforms GRADIEND_Female/Male for BERT base and Distilbert, whereas GRADIEND_Female/Male is better for BERT large and GPT2. The mean proportional rank provides a concise aggregation across all models, offering a robust overview of which methods consistently perform well in debiasing across models.
> - **Minor presentation issues:** We corrected all the mentioned presentation problems:
>     - Explicit phrasing of "Hypothesis (H1)" instead of "(H1)".
>     - Fixing the quotation marks.

---

### Official Review · Reviewer_s9uF · 2025-11-03

**Soundness:** 2
**Presentation:** 3
**Contribution:** 2
**Rating:** 6
**Confidence:** 3

**Summary:**

The paper proposes GRADIEND, a simple encoder–decoder framework that learns interpretable feature neurons from model gradients. It encodes gradient differences tied to sensitive attributes into a scalar value and decodes them into weight updates that can rewrite model parameters to reduce bias. On seven transformer models, GRADIEND is tested to be widely applicable, learning clear feature directions and outperforming baselines when further combined with other post-processing debiasing techniques.

**Strengths:**

1. The idea of encoding feature semantics directly from gradients is elegant, bridging interpretability and parameter-level debiasing with weight-modifying.
2. The proposed method is mathematically well-formulated. The vivid illustration in Fig. 1 and explanations in Section 3 are very clear.
3. I appreciate the experiments. The authors evaluated the method on seven major transformer models covering both encoder- and decoder-only LMs. They conducted a systematic analysis across three bias domains and used multiple bias metrics for thorough benchmarking.
4. The empirical studies demonstrate effective bias modification without major performance loss. The performance can be further boosted when combined with other post-processing techniques.

**Weaknesses:**

1. The encoder–decoder setup might be viewed as a re-parameterization of gradient differences, lacking deeper theoretical justification or analysis of why it works indeed.
2. There is no theoretical analysis of how the task performance would be affected.
3. Binary gender assumption and limited race/religion classes make the fairness conclusions narrow.
4. There is no comparison with more recent causal or reinforcement-based debiasing techniques.

**Questions:**

1. Could you elaborate more on how the encoder can be trained jointly on multiple bias axes?
2. How stable are learned feature neurons across random seeds or pretraining checkpoints?
3. Can the method generalize to multimodal models (e.g., vision–language)?

---

> ### Author Response · Authors · 2025-11-21
> **Answer to Reviewer s9uF**
>
> Thank you for the review. Please see our general comment for a summary of the main revisions made in response to all reviews.
> All references to sections, tables, etc. in our answer correspond to the revised manuscript unless stated otherwise.
>
> **Questions:**
>
> - **Joint Multiple Bias Axes Training:** The main idea is to create a GRADIEND model with multiple feature neurons, one per axis, and use the joint training data of all involved axes. To assist convergence, it may be helpful to initialize weights based on single-axis GRADIENDs. We are already experimenting with this, but early results indicate that training single GRADIENDs for pairs of feature classes (like we do here for race and religion) yields superior performance to joint training. However, this requires a clearly separable axis. We are also exploring truly multi-dimensional features like emotion (e.g., encoded by valence and arousal) and believe that in such a setting, training with multiple neurons will be superior. However, at this time, we do not have the data to support this.
> - **Feature Stability:** We acknowledge that analyzing the stability of the feature neuron is a crucial aspect that we neglected in our current analysis of GRADIEND. We added additional analysis regarding this in the new Appendix E.2, featuring the encoded values distribution of three gender GRADIENDs per base model. Besides the BERT-based models, the encoded values are very similar, with mean sample-based absolute differences of less than 3%. For BERT-base, one training run did not converge to a proper feature encoding. For BERT-large, all three runs converged to proper feature encoding, though one run clearly outperforms the others.
> - **Generalization to Multimodal Models:** While our experiments in this study exclusively focus on MLM and CLM of transformer language models, GRADIEND does not rely on architectural specifics and should apply to any weight-based model trained with gradient-based updates. The only requirement is a feature-related task that provides meaningful gradients along the feature of interest. For a vision-language model, for instance, a gender feature could be extracted by pairing images of humans with known gender labels with texts like "This is a woman/man" in an image-text contrastive learning setting (i.e., CLIP style [1]). Indeed, GRADIEND can be a valuable tool to understand which neurons are used to encode visual information, textual information, and jointly encode information, thereby gaining a better understanding of how early and late fusion mechanisms affect this information encoding.
>
>
> **Weaknesses:**
>
> We acknowledge all weaknesses. At the current time, we have only an intuitive motivation for GRADIEND and no formal derivation for how this works or bounds on required data or similar.
>
> We also acknowledge that the binary view of gender and the three-class perspective of race/religion is limited, though we respectfully submit that these are commonly used data sets and metrics and that this does not automatically invalidate our results.
>
> We further acknowledge that more gender debiasing approaches exist. We tried to select a range of established general debiasing approaches. For instance, Causal-Debias [2]  can only be applied in combination with fine-tuning, and
> [3] does not change model weights, making it less relevant as a comparison to GRADIEND.
>
>
> [1] Radford, A., Kim, J. W., Hallacy, C., Ramesh, A., Goh, G., Agarwal, S., ... & Sutskever, I. (2021, July). Learning transferable visual models from natural language supervision. In International conference on machine learning (pp. 8748-8763). PmLR.
>
> [2] Fan Zhou, Yuzhou Mao, Liu Yu, Yi Yang, and Ting Zhong. 2023. Causal-Debias: Unifying Debiasing in Pretrained Language Models and Fine-tuning via Causal Invariant Learning. In Proceedings of the 61st Annual Meeting of the Association for Computational Linguistics (Volume 1: Long Papers)
>
> [3] Yuchen Cai, Ding Cao, Rongxi Guo, Yaqin Wen, Guiquan Liu, and Enhong Chen. 2024. Locating and Mitigating Gender Bias in Large Language Models. In Advanced Intelligent Computing Technology and Applications: 20th International Conference, ICIC 2024, Tianjin, China, August 5–8, 2024, Proceedings, Part IV. Springer-Verlag, Berlin, Heidelberg, 471–482. https://doi.org/10.1007/978-981-97-5672-8_40

---

### Author Response · Authors · 2025-11-21
**Answer to all Reviewers**

We thank the reviewers for their reviews. Based on all reviews, we have updated the manuscript. The main changes are:

- A stability analysis of the encoded feature neurons across multiple training runs (Appendix E.2).
- An ablation study examining how GRADIEND can be combined with fine-tuning (i.e., applying GRADIEND before and/or after fine-tuning; see Appendix H).
- Additional clarifications and minor details in Sections 2.1, 4, and 6, as well as in Appendices D.2, and G.2

To facilitate review, the supplemental material includes a PDF in which all changes in this revision are marked in blue.
All references to sections, tables, etc. in our answer correspond to the revised manuscript unless stated otherwise.

---

### Author Response · Authors · 2025-12-02
**Final Comment to AC and Reviewers**

Dear AC and Reviewers,

Thank you for the feedback. Below, we concisely summarize our main contributions and the updates made since the initial submission.


**Our Contributions:**

- We introduce **GRADIEND**, a new xAI method for learning a feature of a desired interpretation. All reviewers belive that GRADIEND is a very valuable contribution, calling it, e.g., "elegant", "mathematically well-formulated" (s9uF), "in interesting idea" (jNB4), and "highly effective", "innovative application", and "particularly nuanced and valuable contribution" (43sP).
- We show that GRADIEND reliably learns interpretable features for **gender**, **race**, and **religion** across **seven transformer models** (BERT base/large, DistilBERT, RoBERTa, GPT-2, LLaMA, LLaMA-Instruct), see Section 5.2. The reviewers judge this as "a wide range of experiments [...] on a vast set of models" (jNB4), and "demonstrate that this approach is robust" (43sP).
- We show that GRADIEND enables **behavior modification** (through debiasing measured by SS and SEAT) while preserving language modeling capabilities (GLUE, SuperGLUE), see Section 5.4. Two reviewers directly address this as a strength, due to the "effective bias modification without major performance loss" (s9uF) that has "a high impact" (jNB4). One reviewer further highlights that such work on biases is "profoundly meaningful" and "essential for promoting responsible AI development" (43sP).
- For gender debiasing, GRADIEND reaches **state-of-the-art** among weight-changing approaches (Table 2).
- We report **all raw results** in a comprehensive appendix (e.g., Figures 8-15, Tables 8, 9, 14-29), including bootstrap intervals.
- We provide detailed data descriptions (Appendix B) for all our **eight datasets** created for this work.
- We release **all datasets**, **GRADIEND-modified gender models**, and **code** to enable reproducability of our results.
- Additional analyses including an **overfitting study** (Appendix I) and **examples** (Appendix J).
- In response to reviewer requests, we added **three additional** ablation experiments within the short rebuttal period (see below).

**Changes since Release of Reviews:**

 All changes relative to the original submission are highlighted in blue in a separate version of the manuscript provided in the supplementary material. These changes add additional insights into our methods as suggested by the reviewers and address concerns with missing details. For more details regarding how exactly this addresses individual aspects identified by the reviewers, we refer directly to the responses below.

- **New Appendix E.2:** Stability of GRADIEND across random seeds (Figure 6, Table 7)
- **New Appendix E.3:** Generalization of the feature neuron (Figure 7, dataset description in Appendix B.7)
- **New Appendix H:** Combining GRADIEND with fine-tuning (Table 13)
- Expanded explanations in **Appendix D.2** (multi-token targets) and **Appendix G.2** (proportional rank).
- **Minor clarifications** in the main paper, i.e.,
    - **Section 2.1:** additional comparison SAE (single layer) vs. GRADIEND (all layers)
    - **Section 4:** additional details on generation for race and religion data
    - **Section 5.1:** clarification that we used all model parameters for GRADIEND except for token prediction heads
    - **Section 6:** additional open question (more feature neurons could improve debiasing) and proposal on how to apply CLM with multiple target tokens
    - References to hypotheses are now written as "hypothesis (H1)" instead of just "(H1)"
- **Clarification on the reported language modeling scores in Table 2:** A few debiasing variants (CDA and Dropout-based) are not available for the LLaMA-based models. This makes the reported *absolute* language modeling means per debiasing variant (i.e., averaged across base models) not directly comparable. For this study, however, what matters is the *relative* change compared to each base model, which we already computed and reported as the *average of model-wise differences*. This is the correct quantity for assessing whether a variant harms language modeling performance, even when different sets of base models are used. A reader might instead expect the **difference between averaged scores**, which can appear inconsistent. To avoid misunderstanding, we added a clarification in the **caption of Table 2** and computation details in **Appendix G.2**.

---

### Meta-Review · Area_Chair_NZE9 · 2026-01-10

**Summary:**

The paper is appreciated but also criticised in some aspects. The overall scoring is 6, 4, 8.

Specific questions regard the request of clarification of some punctual parts of the method, while weaknesses are instead more general, addressing the lack of theoretical justification/analysis, bias cases narrow in scope (but indeed common in literature), and missing comparisons with causal or RL-based debiasing methods.

Another set of comments is raised by the most negative reviewer (scoring 4, jNB4) who, while appreciating the novelty of the work, complained about the presentation and the difficulty to well understand the methodology; this is also valid for the experimental part. In addition, there are doubts about the proper functioning for generative models, and again, he raised issues on the evaluation of the results.

No discussion was raised by the above reviewers, who remained silent.

The most positive reviewer (43sP) instead quoted very strong and convincing strengths, and raised weaknesses focused on fine-tuning, which seems to be an aspect overlooked in the paper, asking for comments and clarifications. Specific questions still point to missing method’s details, weak analyses and assessments in the bottleneck layer, as well as insufficient ablations. A discussion was engaged asking for more clarifications, which are provided by the authors.

**Reviewer Concerns:**

The rebuttals to each review are reasonable (not outstanding), but simply acknowledged some of the raised weaknesses and questions. Issues such as, eg., training along multiple bias axes, feature stability, extension to MM models, do not seem to be satisfactorily addressed.
Authors did not provide additional examples (eg., for multiple tokens), and not all further requested results seem to reach SotA.

A discussion was engaged with a Reviewer 43sP (but not with the other two), and more results and explanations are provided, but no further replies from this reviewer are provided.

In general, I cannot say if the rebuttal is fully convincing and satisfactory for all.

**Reviewer Scores:**

Difficult to say if the first 2 reviewers (s9uF, jNB4) are satisfied of the authors' feedback (no discussion), and even the third one (43sP) did not reply further after the last answers provided by the authors'.

Nevertheless, the summary of the changes reported by authors underline the revision made and the new material inserted, which is convincing, and, since 2 out of 3 reviewers were originally positive, I would maintain the relative ranking.

---

### Decision · Program_Chairs · 2026-01-26

Accept (Poster)